

# Finite-size corrections in critical symmetry-resolved entanglement

**Benoit Estienne[1], Yacine Ikhlef[1] and Alexi Morin-Duchesne[2]**

**1** Sorbonne Université, CNRS, Laboratoire de Physique Théorique et Hautes Énergies,
LPTHE, F-75005 Paris, France
**2** Max Planck Institut für Mathematik, 53111 Bonn, Germany

## Abstract

In the presence of a conserved quantity, symmetry-resolved entanglement entropies are a refinement of the usual notion of entanglement entropy of a subsystem. For critical 1d quantum systems, it was recently shown in various contexts that these quantities generally obey *entropy equipartition* in the scaling limit, i.e. they become independent of the symmetry sector. In this paper, we examine the finite-size corrections to the entropy equipartition phenomenon, and show that the nature of the symmetry group plays a crucial role. In the case of a discrete symmetry group, the corrections decay algebraically with system size, with exponents related to the operators' scaling dimensions. In contrast, in the case of a U(1) symmetry group, the corrections only decay *logarithmically* with system size, with model-dependent prefactors. We show that the determination of these prefactors boils down to the computation of twisted overlaps.

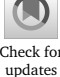

# 1   Introduction

Entanglement plays a prominent role in the study of quantum many-body physics. For instance for one-dimensional systems, a well-controlled entanglement is a necessary condition for the efficiency of density matrix renormalization group and matrix-product state approaches. Given a bipartition $A \cup B$ of a quantum system, the entanglement between $A$ and $B$ in a state $|\psi\rangle$ is encoded in the reduced density matrix $\rho_A = \text{Tr}_B |\psi\rangle\langle\psi|$. A common way of quantifying entanglement is to consider the von Neumann and Rényi entropies:

$$S(A) = -\text{Tr}_A(\rho_A \log \rho_A), \qquad S_n(A) = \frac{1}{1-n} \text{Tr}_A(\rho_A^n).$$

Over the last decade, following [1], it was also realised that the full spectrum of $\rho_A$, called the *entanglement spectrum*, also contains relevant information, especially for the understanding of topological order. This entanglement spectrum may be derived from the knowledge of all Rényi entropies $S_n(A)$ for positive integer $n$, through careful analysis [2]. Alternatively, for a 1+1d critical system described by a Conformal Field Theory (CFT), the entanglement spectrum can be obtained from the same CFT on the infinite strip with appropriate boundary conditions [3].

As argued in [4], when an additive symmetry (or conserved charge) is present, one may define an intermediary description of entanglement, by considering the *symmetry-resolved* entanglement entropies, which are essentially the analogs of von Neumann and Rényi entropies for the blocks of the reduced density matrix $\rho_A$ associated to the values of the local charge. Examples include the local magnetisation for a spin system, the number of particles for a quantum gas, *etc*. It was later realised that, for 1+1d critical systems with U(1) symmetry, the (properly normalised) symmetry-resolved entropies generally satisfy an equipartition property [5] in the scaling limit, *i.e.* over a large range of values of the local charge, the symmetry-resolved entropies are independent of the charge. The study of symmetry-resolved entanglement entropies has been applied to a variety of situations [6–17]. For quantum computing, these entropies also serve as building blocks for the definition of operationally accessible entanglement [18, 19]. The problem of finite-size scaling for entanglement entropies has also been addressed [20–22], and in particular some unconventional corrections to scaling, originating from subdominant twist operators, have been identified [22].

In the present paper, we are concerned with the finite-size corrections to equipartition of symmetry-resolved entropies for critical 1+1d systems. We employ essentially the technique introduced in [6], which relies on the "replica trick" [23] or cyclic orbifold [24] formulation, and more specifically on the introduction of composite twist operators inserting flux lines (which are nothing but topological defects) conjugate to the local charge, ending at the branch points connecting the replicas. We take specifically into account the variations of the non-

universal normalisation of lattice operators, and show that they can give rise to significant finite-size corrections.

In fact, this effect can be illustrated on a simple example in the critical XXZ spin chain, by looking at the probability distribution $p(S_A^z)$ of the magnetisation of subregion $A$, which can be viewed itself as a measure of entanglement [4]. Consider for simplicity the case when $A$ is a single interval of length $\ell$ in an infinite chain. For a very large interval, it is a standard result of bosonisation that the fluctuations of $S_A^z$ are Gaussian. Indeed, since the generating function $\widehat{Z}(\alpha)$ of $p(S_A^z)$ is the two-point function of lattice vertex operators with scaling dimension $\kappa\alpha^2/4$, where $\kappa$ is a universal constant, it behaves in the scaling limit as $\widehat{Z}(\alpha) \propto \ell^{-\kappa\alpha^2/2}$, and hence, by inverse Fourier transform, one gets for $p(S_A^z)$ a Gaussian distribution with a variance $\sim \sqrt{\kappa\log\ell}$. Now, to refine the argument, one should be aware that the prefactor in $\widehat{Z}(\alpha)$, originating from the normalisation of lattice operators, although it does not depend on $\ell$, has a non-trivial dependence on the vertex charge $\alpha$. Hence, one should write

$$\widehat{Z}(\alpha) = |\mathcal{A}(\alpha)|^2 \, \ell^{-\kappa\alpha^2/2}, \qquad |\mathcal{A}(\alpha)|^2 = 1 + a_1\alpha^2 + a_2\alpha^4 + \dots,$$

where the coefficients $a_1, a_2, \dots$ are non-universal constants. After simple manipulations (calculation details are given in Sec. 3), one finds the corrected probability density for the rescaled magnetisation $x = S_A^z/\sqrt{\kappa\log\ell}$ in the form:

$$\rho(x) \simeq \frac{\exp(-x^2/2)}{\sqrt{2\pi}} \times \left[1 + \frac{a_1 q_1(x)}{\kappa\log\ell} + \frac{a_2 q_2(x)}{(\kappa\log\ell)^2} + \frac{a_3 q_3(x)}{(\kappa\log\ell)^3} + \dots\right],$$

where $a_1, a_2, a_3, \dots$ are the series expansion coefficients of $|\mathcal{A}(\alpha)|^2$ discussed above, whereas $q_1(x)$, $q_2(x)$, $q_3(x), \dots$ are universal polynomial functions derived from the moments of a normal distribution. Hence the variations of the normalisation factor give rise to logarithmic corrections to the Gaussian distribution of $S_A^z$. This simple line of arguments extends to symmetry-resolved entropies, as we shall explain in Sec. 3.

The layout of the paper is as follows. In Sec. 2, we review the definitions of symmetry-resolved entanglement entropies, and the approach based on cyclic orbifolds and composite twist operators inserting both a branch point and a flux line conjugate to the conserved charge. In Sec. 3, we consider the entanglement of a single interval $A$ in the critical XXZ spin chain: we first compute the finite-size corrections for the magnetisation distribution of $A$, and then for the symmetry-resolved Rényi entropies. For both types of quantities, we show that these corrections decay logarithmically, as sketched above. We then derive the relation between the normalisation factors of lattice twist operators and the scalar product between eigenstates of the Hamiltonian with appropriate twisted boundary conditions. Through this approach, we compute explicitly for the XX case the normalisation factors for composite twist operators, by using Wick's theorem. In Sec. 4, we consider the critical $\mathbb{Z}_p$ clock models [25], and show that, even when normalisation factors are taken into account, the corrections to the equipartition rule for resolved entropies associated to the $\mathbb{Z}_p$ symmetry remain algebraic, with a very simple dependence on the symmetry sector. In Sec. 5, we conclude with final remarks and perspectives. The calculation details for the XX case are given in the Appendix.

# 2 Symmetry-resolved entropies

## 2.1 Definitions

Let us briefly recall the general definition of symmetry-resolved entropies. Consider a quantum model with an internal (continuous or discrete) symmetry, as described by a conserved charge $Q$. The total charge $Q$ is assumed to be a sum of local operators: upon partitioning the system into two spatial subregions $\mathcal{S} = A \cup B$, it can be split as

$$Q = Q_A + Q_B \,, \tag{2.1}$$

where $Q_A$ and $Q_B$ are Hermitian operators acting only on the degrees of freedom of $A$ and $B$, respectively. At equilibrium, the state of the total system $\mathcal{S}$ is described by a density matrix $\rho$ that commutes with $Q$, and in turn the reduced density matrix $\rho_A = \mathrm{Tr}_B(\rho)$ of the subsystem $A$ commutes with $Q_A$. This means that the subsystem $A$ is described by a statistical ensemble of different charge sectors, in the sense that its reduced density matrix is block-diagonal with respect to the eigenspaces of $Q_A$:

$$\rho_A = \sum_q p_q \, \rho_A(q) \,, \qquad \mathrm{Tr}_A[\rho_A(q)] = 1 \,, \qquad Q_A \rho_A(q) = q \, \rho_A(q) \,. \tag{2.2}$$

When measuring the charge in region A, *i.e.* the observable $Q_A$, the outcome $q$ is obtained with probability $p_q$. After such a measurement, the reduced density matrix describing the subsystem $A$ collapses to $\rho_A(q)$. In terms of the orthogonal projector $\Pi_q$ onto the subspace $Q_A = q$, we have

$$p_q = \mathrm{Tr}_A(\Pi_q \rho_A) \,, \qquad \rho_A(q) = \frac{\Pi_q \rho_A \Pi_q}{p_q} \,, \tag{2.3}$$

provided that $p_q \neq 0$.

Symmetry-revolved entropies are obtained by using the above decomposition to refine the usual notion of entanglement entropy. Inserting the decomposition $\rho_A = \sum_q p_q \rho_A(q)$ over the $Q_A$ sectors in the Von Neumann entanglement entropy

$$S_{\mathrm{vN}} := -\mathrm{Tr}_A(\rho_A \log \rho_A) \,, \tag{2.4}$$

one is lead to the following decomposition [5]

$$S_{\mathrm{vN}} = \sum_q p_q S_{\mathrm{vN}}(q) - \sum_q p_q \log p_q \,, \tag{2.5}$$

where $S_{\mathrm{vN}}(q)$ is the so-called *symmetry-resolved (Von Neumann) entanglement entropy*

$$S_{\mathrm{vN}}(q) := -\mathrm{Tr}_A[\rho_A(q) \log \rho_A(q)] \,. \tag{2.6}$$

Thus there are two contributions to the total entanglement entropy of region $A$. The first one is just the average of the entanglement entropies coming from each $Q_A$ sector, while the second one comes from the charge fluctuations.

In order to compute the Von Neumann entanglement entropy, one often resorts to the replica trick, thus introducing $n$ copies of the system. In the present context, the $n^{\mathrm{th}}$ symmetry-resolved Renyi entropy is defined as

$$S_n(q) := \frac{1}{1-n} \log \mathrm{Tr}_A[\rho_A^n(q)] \,. \tag{2.7}$$

Alternatively, we can introduce the partition functions

$$Z_n(q) := \mathrm{Tr}_A(\Pi_q \, \rho_A^n), \tag{2.8}$$

and write the quantities of interest as

$$p_q = Z_1(q), \qquad S_n(q) = \frac{1}{1-n} \left[ \log Z_n(q) - n \log Z_1(q) \right]. \tag{2.9}$$

The symmetry-resolved Von Neumann entropy is then obtained as

$$S_{\mathrm{vN}}(q) = -\frac{d}{dn} \left[ \frac{Z_n(q)}{Z_1(q)^n} \right]_{n=1}. \tag{2.10}$$

Thus computing the entanglement entropy boils down to computing the partition functions $Z_n(q)$. As was observed in [5,6], it is convenient to introduce the twisted partition functions

$$\widehat{Z}_n(\alpha) := \mathrm{Tr}_A\left(e^{i\alpha Q_A} \rho_A^n\right) = \sum_q e^{i\alpha q} Z_n(q). \tag{2.11}$$

In particular, $\widehat{Z}_1(\alpha)$ is the generating function for the probabilities $p_q$. The generating functions $\widehat{Z}_n(\alpha)$ also appear in a different context under the name of *charged Renyi entropies* [26–33].

## 2.2 Cyclic orbifold formalism

For any integer $n > 1$, starting from any discrete or continuous model $\mathcal{M}$, the $\mathbb{Z}_n$ cyclic orbifold is defined as the model consisting of $n$ decoupled copies $\mathcal{M}$, where any $n$-uplet of configurations is identified with its image under any cyclic permutation. We can write symbolically:

$$\mathrm{Orb}_n(\mathcal{M}) = (\underbrace{\mathcal{M} \times \cdots \times \mathcal{M}}_{n})/\mathbb{Z}_n. \tag{2.12}$$

If $A$ consists of $p$ intervals $A = [x_1, x_2] \cup \cdots \cup [x_{2p-1}, x_{2p}]$, then $\widehat{Z}_n(\alpha)$ is given by the $2p$-point function, in the orbifold $\mathrm{Orb}_n(\mathcal{M})$, of a composite twist operator $\tau_\alpha$:

$$\widehat{Z}_n(\alpha) \propto \langle \tau_\alpha^\dagger(x_1) \tau_\alpha(x_2) \ldots \tau_\alpha^\dagger(x_{2p-1}) \tau_\alpha(x_{2p}) \rangle. \tag{2.13}$$

In the scaling limit, the twist operator $\tau_\alpha(x)$ is obtained as the most relevant term in the OPE $\tau \cdot \mathcal{O}_\alpha$, where $\tau$ is the bare twist operator implementing a branch point, and $\mathcal{O}_\alpha$ is the "disorder operator" associated to the conservation of $Q$. This construction will be made more explicit in the examples studied below. The conformal dimensions of $\tau$ and $\tau_\alpha$ are:

$$h_\tau = \frac{c}{24}\left(n - \frac{1}{n}\right), \qquad h_{\tau_\alpha} = h_\tau + \frac{h_\alpha}{n}, \tag{2.14}$$

where $h_\alpha$ is the conformal dimension of $\mathcal{O}_\alpha$.

In this paper, we will restrict to the single-interval entropies, with $A = [0, \ell]$, in a 1d critical quantum chain of length $L$ with periodic boundary conditions, at inverse temperature $\beta$. We do not consider the case in which both $L$ and $\beta$ are finite, as it involves the two-point function of $\tau_\alpha$ on a torus. The generating function $\widehat{Z}_n(\alpha)$ then relates to the two-point function of twist operators:

$$\langle \tau_\alpha^\dagger(0)\tau_\alpha(\ell) \rangle = \begin{cases} \ell^{-4h_{\tau_\alpha}} & \text{for } L = \infty \text{ and } \beta = \infty\,, \\ \left(\frac{L}{\pi}\sin\frac{\pi\ell}{L}\right)^{-4h_{\tau_\alpha}} & \text{for finite } L \text{ and } \beta = \infty\,, \\ \left(\frac{\beta}{\pi}\sinh\frac{\pi\ell}{\beta}\right)^{-4h_{\tau_\alpha}} & \text{for } L = \infty \text{ and finite } \beta\,. \end{cases} \tag{2.15}$$

We will focus our attention on the asymptotic behaviour of the resolved entropies in the scaling limit, by taking into account the finite-size corrections to (2.15) due not only to subdominant operators, but also to the normalisation of lattice operators.

In the next sections, we shall develop the arguments and numerical computations for the case of finite $L$ and $\beta = \infty$, but they can be easily extended to the other cases.

## 3 The XXZ spin-1/2 chain

We consider the XXZ Hamiltonian on $N$ sites defined by the Pauli matrices $\sigma^{x,y,z}$, and acting on the Hilbert space $(\mathbb{C}^2)^{\otimes N}$:

$$\mathcal{H}_{\text{XXZ}} := -\frac{1}{2}\sum_{j=1}^{N}\left[\sigma_j^x\sigma_{j+1}^x + \sigma_j^y\sigma_{j+1}^y + \Delta(\sigma_j^z\sigma_{j+1}^z - 1)\right], \qquad \begin{cases} \Delta = -\cos\gamma, \\ 0 < \gamma < \pi, \end{cases} \tag{3.1}$$

with periodic boundary conditions $\sigma_{N+1}^{x,y,z} := \sigma_1^{x,y,z}$. For simplicity, we consider only the case of even $N$. The total magnetisation is a local conserved charge:

$$S^z = \frac{1}{2}\sum_{j=1}^{N}\sigma_j^z, \qquad [\mathcal{H}_{\text{XXZ}}, S^z] = 0. \tag{3.2}$$

The group $U(1)$ is then realised by the operators $e^{i\alpha S^z}$ with $\alpha \in \mathbb{R}/2\pi\mathbb{Z}$. Consider the partition of the lattice sites into two subsystems $A = \{1, 2, \ldots, r\}$ and $B = \{r+1, \ldots, N\}$. The magnetisation of subsystem $A$ is

$$q = S_A^z = \frac{1}{2}\sum_{j=1}^{r}\sigma_j^z, \tag{3.3}$$

and it has eigenvalues $-r/2, -r/2 + 1, \ldots, r/2$. The generating function of resolved entropies on the lattice is given by

$$\widehat{Z}_n(\alpha) = \text{Tr}_A\left(e^{i\alpha S_A^z}\rho_A^n\right), \tag{3.4}$$

where $-\pi < \alpha < \pi$, and $\rho_A$ is the reduced density matrix. One recovers the functions $Z_n(q)$ as

$$Z_n(q) = \frac{1}{2\pi}\int_{-\pi}^{+\pi}d\alpha\, e^{-i\alpha q}\,\widehat{Z}_n(\alpha), \qquad q = -\frac{r}{2}, -\frac{r}{2} + 1, \ldots, \frac{r}{2}. \tag{3.5}$$

Note that this relation holds for any length $r$, even or odd.

### 3.1 Probability distribution of the partial magnetisation

Let us begin with the case $n = 1$, i.e. the probability distribution of the partial magnetisation $q = S_A^z$:

$$p_q = Z_1(q) = \frac{1}{2\pi}\int_{-\pi}^{+\pi}d\alpha\, e^{-i\alpha q}\,\widehat{Z}_1(\alpha). \tag{3.6}$$

For $n = 1$, the generating function $\widehat{Z}_1(\alpha)$ is simply a groundstate expectation value:

$$\widehat{Z}_1(\alpha) = \langle\psi|e^{i\alpha S_A^z}|\psi\rangle = \langle\psi|\prod_{j=1}^{r}e^{i\alpha\sigma_j^z/2}|\psi\rangle. \tag{3.7}$$

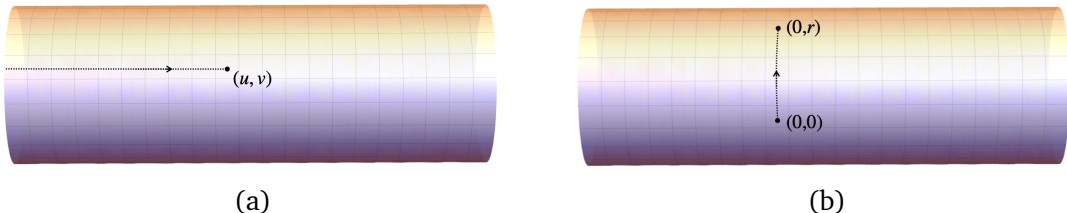

Figure 1: (a) The operator $\mathbf{v}_\alpha(u,v)$ in the six-vertex model. (b) The two-point function $\langle \mathbf{v}_\alpha^\dagger(0,0)\mathbf{v}_\alpha(0,r)\rangle$. Each oriented dotted line represents the product of operators $\exp(i\alpha\sigma_j^z/2)$ on the edges it crosses.

In terms of the six-vertex (6V) model associated to the XXZ chain, this becomes a two-point function on the infinite cylinder of circumference $N$ sites (see Fig. 1):[1]

$$\widehat{Z}_1(\alpha) = \langle \mathbf{v}_\alpha^\dagger(0,0)\mathbf{v}_\alpha(0,r)\rangle = \langle \mathbf{v}_{-\alpha}(0,0)\mathbf{v}_\alpha(0,r)\rangle \,, \tag{3.8}$$

where $\mathbf{v}_\alpha(u,v)$ inserts a chain of operators $e^{i\alpha\sigma^z/2}$ along a path on the dual lattice, going from $(-\infty,0)$ to $(u,v)$. Since the 6V $R$-matrix commutes with $(e^{i\alpha\sigma^z/2}\otimes e^{i\alpha\sigma^z/2})$, the lattice operators $\mathbf{v}_\alpha$ are mutually local, *i.e.* their correlation functions are independent of the choice of the paths.

In the thermodynamic limit, the universal behavior of the 6V model is captured by a compact boson CFT with central charge $c = 1$, where a family of primary operators is given by the vertex operators $V_\beta$, with scaling dimensions

$$h_\beta = \bar{h}_\beta = \frac{(\beta/2\pi)^2}{4g} \,, \qquad g = \frac{\pi-\gamma}{\pi} \,, \qquad \Delta = \cos\pi g \,. \tag{3.9}$$

By convention, we normalise the $V_\beta$'s so that the two-point function reads

$$\langle V_\beta(z,\bar{z})V_{-\beta}(w,\bar{w})\rangle = \frac{1}{|z-w|^{4h_\beta}} \,. \tag{3.10}$$

Back to the 6V model, if we fix $\alpha$ in the interval $-\pi < \alpha < \pi$, the lattice operator $\mathbf{v}_\alpha$ is described in the scaling limit [34, 35] by the vertex operator $V_\alpha$:

$$\mathbf{v}_\alpha(u,v) \simeq \mathcal{A}_1(\alpha)a^{2h_\alpha}V_\alpha(z,\bar{z}), \tag{3.11}$$

where $a$ is the lattice spacing, and $z = a(u+iv), \bar{z} = a(u+iv)$ with integers $u$ and $v$. The prefactor $\mathcal{A}_1(\alpha) = \overline{\mathcal{A}_1(-\alpha)}$ is independent of system size but non-universal, except for $\mathcal{A}_1(0) = 1$, since $\mathbf{v}_0$ is simply the identity operator on the lattice. For the two-point function in particular, this means

$$\widehat{Z}_1(\alpha) \simeq |\mathcal{A}_1(\alpha)|^2 \, a^{4h_\alpha} \, \langle V_{-\alpha}(0,0)V_\alpha(0,r)\rangle = |\mathcal{A}_1(\alpha)|^2 \left(\frac{L}{\pi a}\sin\frac{\pi\ell}{L}\right)^{-4h_\alpha}, \tag{3.12}$$

where

$$L = aN \,, \qquad \ell = ar \,. \tag{3.13}$$

---

[1] In this article, the distinction between lattice operators and the CFT operators describing their scaling limits plays an important role. For this reason, we denote lattice operators with lowercase bold letters such as $\mathbf{v}$ and $\mathbf{t}$.

However it should be stressed that in (3.11) and (3.12) the *r.h.s.* is only the most relevant term (as $a \to 0$), and there are also contributions coming from less relevant scaling fields (generically all fields with compatible quantum numbers) with U(1) charges $\alpha, \alpha \pm 2\pi, \alpha \pm 4\pi \ldots$ This means all the vertex operators and descendants, with a U(1) charge equal to $\alpha$ modulo $2\pi$. We write this as

$$\mathbf{v}_\alpha(u,v) = \sum_{m \in \mathbb{Z}} (-1)^{m(u+v)} \mathcal{A}_1(\alpha + 2\pi m) \, a^{2h_{\alpha+2\pi m}} \, V_{\alpha+2\pi m}(z, \bar{z}) + \text{desc.}, \qquad (3.14)$$

where "desc." denotes the contribution from the descendant operators. The sign $(-1)^{u+v}$ appears for each contribution of $V_{\alpha+2\pi m}$ with $m$ odd, reflecting the fact that the transfer-matrix eigenstate associated to $V_{\alpha+2\pi m}$ has momentum $m\pi$. From this expansion of $\mathbf{v}_\alpha$, we get for any $\alpha$ in the interval $-\pi < \alpha < \pi$:

$$\widehat{Z}_1(\alpha) \simeq \sum_{m=-\infty}^{\infty} (-1)^{mr} \, |\mathcal{A}_1(\alpha + 2\pi m)|^2 \left( \frac{L}{\pi a} \sin \frac{\pi \ell}{L} \right)^{-4h_{\alpha+2\pi m}} + \text{desc.} \qquad (3.15)$$

The corrections to (3.12) are the usual finite-size correction for a critical lattice model. In particular they decay algebraically with system size (taking $L, \ell \to \infty$ with $\ell/L$ finite).

But as it turns out, the probability distribution of the partial magnetisation has much larger finite-size corrections that only decay logarithmically. These originate from the $\alpha$-dependence of the prefactor $\mathcal{A}_1(\alpha)$. Indeed up to algebraically subleading terms, we have

$$\widehat{Z}_1(\alpha) \simeq |\mathcal{A}_1(\alpha)|^2 \left( \frac{L}{\pi a} \sin \frac{\pi \ell}{L} \right)^{-4h_\alpha} = |\mathcal{A}_1(\alpha)|^2 \, e^{-K_\ell \alpha^2/2}, \qquad (3.16)$$

where we have introduced the short-hand notation adapted to the case of finite $L$ and zero temperature:

$$K_\ell = \frac{1}{2\pi^2 g} \log \left( \frac{L}{\pi a} \sin \frac{\pi \ell}{L} \right). \qquad (3.17)$$

If we write the series expansion of $\log |\mathcal{A}_1(\alpha)|^2$ as

$$\log |\mathcal{A}_1(\alpha)|^2 = \sum_{j=1}^{\infty} \frac{b_j}{(2j)!} \alpha^{2j}, \qquad (3.18)$$

we see that the $2j$-th cumulant of the probability distribution $p_q$ is simply given by the coefficient $(-1)^j b_j$ for $j > 1$, whereas the variance is $(K_\ell - b_1)$. All odd cumulants are zero. Recall that the mean value of $q$ vanishes exactly, due to symmetry under spin reversal.

Let us write more explicitly the probability distribution. Using the relation (3.6), one gets

$$\begin{aligned} p_q &\simeq \frac{1}{2\pi} \int_{-\pi}^{+\pi} d\alpha \, |\mathcal{A}_1(\alpha)|^2 \exp\left[ -i\alpha q - \frac{K_\ell \alpha^2}{2} \right] \\ &\simeq \frac{e^{-q^2/2K_\ell}}{2\pi} \int_{-\pi}^{+\pi} d\alpha \, |\mathcal{A}_1(\alpha)|^2 \exp\left[ -\frac{K_\ell}{2} \left( \alpha + i\frac{q}{K_\ell} \right)^2 \right]. \end{aligned} \qquad (3.19)$$

Interestingly, we note that if we include the contributions from the subleading vertex operators $V_{\alpha+2\pi m}$ with $m \neq 0$, everything adds up nicely to

$$p_q \simeq \frac{e^{-q^2/2K_\ell}}{2\pi} \int_{-\infty}^{+\infty} d\alpha \, |\mathcal{A}_1(\alpha)|^2 \exp\left[ -\frac{K_\ell}{2} \left( \alpha + i\frac{q}{K_\ell} \right)^2 \right]. \qquad (3.20)$$

In particular the signs $(-1)^{mr}$ end up disappearing because $q \in \{-r/2, -r/2 + 1, \ldots, r/2\}$. The relations (3.19) and (3.20) are not incompatible. They both give the correct asymptotic behavior of $p_q$ in the limit $K_\ell \gg 1$. They only differ in subleading corrections that decay exponentially in $K_\ell$, which means algebraically with system size.

Introducing the scaling variable $x = q/\sqrt{K_\ell}$, with probability distribution

$$\rho(x) = \sqrt{K_\ell}\, p_{x\sqrt{K_\ell}}, \tag{3.21}$$

we can write:

$$\rho(x) \simeq \frac{e^{-x^2/2}}{\sqrt{2\pi}} \times \mathcal{I}_1(K_\ell, x), \qquad \mathcal{I}_1(K, x) := \int_{-\infty}^{+\infty} \frac{dw}{\sqrt{2\pi}} \left| \mathcal{A}_1\left(\frac{w}{\sqrt{K}}\right) \right|^2 e^{-\frac{(w+ix)^2}{2}}. \tag{3.22}$$

At leading order, as $K_\ell \to \infty$, we have $\mathcal{I}_1(K_\ell, x) \to 1$, and the variable $x$ follows the expected normal distribution

$$\rho(x) \simeq \frac{e^{-x^2/2}}{\sqrt{2\pi}}. \tag{3.23}$$

But this leading behavior is corrected by corrections that decay *algebraically* with $K_\ell$, thus logarithmically with system size. These corrections are fully controlled by the coefficient $|\mathcal{A}_1(\alpha)|^2$, or more precisely by its Taylor expansion around $\alpha = 0$:

$$|\mathcal{A}_1(\alpha)|^2 = \sum_{j=0}^{\infty} a_j \alpha^{2j}, \qquad a_0 = 1. \tag{3.24}$$

The integral $\mathcal{I}_1(K, x)$ then takes the form of a power series in $1/K$:

$$\mathcal{I}_1(K, x) = \sum_{j=0}^{\infty} \frac{a_j\, q_j(x)}{K^j}, \qquad q_j(x) = \sum_{p=0}^{j} \frac{(2j)!}{(2j-2p)!\,p!} \frac{(-x^2)^{j-p}}{2^p}. \tag{3.25}$$

As a result, we get the following asymptotic behavior for the probability distribution of the rescaled partial magnetisation as $K_\ell \to \infty$:

$$\rho(x) = \frac{e^{-x^2/2}}{\sqrt{2\pi}} \times \left[ 1 + \frac{a_1}{K_\ell}(1 - x^2) + \frac{a_2}{K_\ell^2}(3 - 6x^2 + x^4) + \ldots \right], \tag{3.26}$$

where $K_\ell$ is given in (3.17) and the parameter $0 < g < 1$ is related to the XXZ anisotropy via $\Delta = \cos \pi g$.

While in the above we considered a system of $L$ sites with periodic boundary conditions at zero temperature, the same results holds for an infinite system at inverse temperature $\beta$, with $K_\ell$ replaced by

$$K_\ell \to \frac{1}{2\pi^2 g} \log\left( \frac{\beta}{\pi} \sinh \frac{\pi \ell}{\beta} \right). \tag{3.27}$$

## 3.2 Charge-resolved Rényi entropies

The generalisation to charge-resolved Rényi entropies, *i.e.* to the case $n > 1$, is straightforward. For $\alpha = 0$, the generating function $\widehat{Z}_n(0)$ can be viewed as the partition function of the 6V model on the replicated cylinder with a branch cut connecting the copies $\mu$ and $(\mu+1) \mod n$ across the interval, or, equivalently, as the correlation function

$$\widehat{Z}_n(0) = \langle \mathbf{t}^\dagger(0,0)\mathbf{t}(0,r)\rangle, \tag{3.28}$$

where $\mathbf{t}(u,v)$ is the lattice twist operator. Computing $\widehat{Z}_n(\alpha)$ for generic $\alpha$ then amounts to inserting $\mathbf{v}_\alpha^\dagger(0,0)\mathbf{v}_\alpha(0,r)$ on a single copy. Introducing the lattice composite twist operator $\mathbf{t}_\alpha(u,v) := \mathbf{t}(u,v)\mathbf{v}_\alpha(u,v)$, we can simply write $\widehat{Z}_n(\alpha)$ as the two-point function

$$\widehat{Z}_n(\alpha) = \langle \mathbf{t}_\alpha^\dagger(0,0)\mathbf{t}_\alpha(0,r)\rangle. \tag{3.29}$$

In the scaling limit, the composite twist operator [24,36,37] is obtained by the point-splitting procedure:

$$\tau_\alpha(z,\bar{z}) = n^{2h_\alpha} \lim_{\varepsilon \to 0} \left[ |\varepsilon|^{2(1-1/n)h_\alpha} V_\alpha(z+\varepsilon,\bar{z}+\bar{\varepsilon})\tau(z,\bar{z})\right], \tag{3.30}$$

and has conformal dimension $h_{\tau_\alpha} = h_\tau + h_\alpha/n$. The factor $n^{2h_\alpha}$ is included to ensure that the two-point function is normalised as in (2.15). Similarly to $\mathbf{v}_\alpha$, the lattice twist operator decomposes as

$$\mathbf{t}_\alpha(u,v) = \sum_{m=-\infty}^{+\infty} (-1)^{m(u+v)} \mathcal{A}_n(\alpha+2\pi m) a^{2h_{\tau_{\alpha+2\pi m}}} \tau_{\alpha+2\pi m}(z,\bar{z}) + \text{desc.}, \tag{3.31}$$

with $z = u+iv, \bar{z} = u-iv$, and where $\mathcal{A}_n(\alpha)$ is a non-universal prefactor independent of the system size. This yields the following decomposition for $\widehat{Z}_n(\alpha)$:

$$\widehat{Z}_n(\alpha) \simeq \sum_{m=-\infty}^{\infty} (-1)^{mr} |\mathcal{A}_n(\alpha+2\pi m)|^2 \left(\frac{L}{\pi a}\sin\frac{\pi\ell}{L}\right)^{-4(h_\tau+h_{\alpha+2\pi m}/n)} + \text{desc.}, \tag{3.32}$$

where again "desc." denotes the contribution from the descendant operators. In particular, up to exponentially small corrections in $K_\ell$, we have

$$\widehat{Z}_n(0) \simeq |\mathcal{A}_n(0)|^2 \times \left(\frac{L}{\pi a}\sin\frac{\pi\ell}{L}\right)^{-4h_\tau}. \tag{3.33}$$

Note that $\widehat{Z}_n(0) = \text{Tr}_A\left(\rho_A^n\right) = Z_n$ is simply the partition function on the replicated surface, which is related to the usual Rényi entropy via

$$S_n = \frac{1}{1-n}\log Z_n. \tag{3.34}$$

Proceeding exactly like for the probability distribution, we get

$$\frac{Z_n(q = x\sqrt{K_\ell})}{Z_n} \simeq \frac{e^{-nx^2/2}}{\sqrt{2\pi K_\ell/n}} \times \mathcal{I}_n(K_\ell/n, x\sqrt{n}), \tag{3.35}$$

where

$$\mathcal{I}_n(K,x) := \int_{-\infty}^{+\infty} \frac{dw}{\sqrt{2\pi}} \left|\frac{\mathcal{A}_n(w/\sqrt{K})}{\mathcal{A}_n(0)}\right|^2 e^{-\frac{(w+ix)^2}{2}}. \tag{3.36}$$

The integral $\mathcal{I}_n(K, x)$ can be expanded for $K \to \infty$ as

$$\mathcal{I}_n(K, x) = \sum_{j=0}^{\infty} \frac{a_{n,j} q_j(x)}{K^j} = 1 + \frac{a_{n,1}}{K}(1 - x^2) + \frac{a_{n,2}}{K^2}(3 - 6x^2 + x^4) + \dots, \tag{3.37}$$

where the polynomials $q_j(x)$ are given in (3.25), and the $a_{n,j}$'s are the Taylor coefficients of $\mathcal{A}_n^2(\alpha)/\mathcal{A}_n^2(0)$:

$$\left| \frac{\mathcal{A}_n(\alpha)}{\mathcal{A}_n(0)} \right|^2 = \sum_{j=0}^{\infty} a_{n,j}\, \alpha^{2j}\,. \tag{3.38}$$

As a result, we can write the charge-resolved entropies as:

$$S_n(q) = S_n - \frac{1}{2}\log(2\pi K_\ell) + \frac{\log n}{2(1-n)} + c_n(K_\ell, q/\sqrt{K_\ell}), \tag{3.39}$$

up to terms exponentially small in $K_\ell$. Hence, in finite size, the leading corrections to equipartition decay algebraically with $K_\ell$, and are encoded in the function

$$c_n(K, x) = \frac{1}{1-n}\left[ \log \mathcal{I}_n(K/n, x\sqrt{n}) - n \log \mathcal{I}_1(K, x) \right]. \tag{3.40}$$

In the limit $K \to \infty$, the dominant term in $c_n$ is

$$c_n(K, x) \simeq \frac{n}{K}(A_n - B_n x^2), \tag{3.41}$$

where $A_n = (a_{n,1} - a_1)/(1-n)$ and $B_n = (n a_{n,1} - a_1)/(1-n)$. Hence, $c_n$ is of order $1/\log(L/a)$, and the charge-resolved entropies become equipartitioned in the scaling limit, but up to very large logarithmic corrections encoded in the function $c_n(K, x)$. These corrections are model-dependent, because they originate from the prefactors $\mathcal{A}_n(\alpha)$ in the lattice twist operator $\mathbf{t}_\alpha$.

## 3.3 Lattice prefactors and scalar products

Although the non-universal prefactors $\mathcal{A}_n(\alpha)$ cannot be computed by CFT methods, in this section we shall show that they are simply related to scalar products of XXZ eigenstates. Underlying this relation is simply the state-operator correspondence. The relation is generic and holds for any lattice model with a local symmetry, but for the reader's convenience we illustrate it on the XXZ spin chain, in a way that should be straightforward to translate to other models.

We first expose the argument for $n = 1$, *i.e.* for the normalisation factor $\mathcal{A}_1(\alpha)$ associated to the lattice operator $\mathbf{v}_\alpha$. Recall that if $(u, v)$ and $(u', v')$ are two lattice points on the cylinder, then we have, for $-\pi < \alpha < \pi$,

$$\langle \mathbf{v}_\alpha^\dagger(u, v)\mathbf{v}_\alpha(u', v') \rangle \simeq |\mathcal{A}_1(\alpha)|^2\, a^{4h_\alpha} \left\langle V_{-\alpha}(z)V_\alpha(z') \right\rangle = \frac{|\mathcal{A}_1(\alpha)|^2}{\left| \frac{L}{\pi a} \sinh\frac{\pi}{L}(z - z') \right|^{4h_\alpha}}, \tag{3.42}$$

where $z = a(u + iv)$ and $z' = a(u' + iv')$. In particular setting $(u, v) = (0, 0)$ and $(u', v') = (-M, 0)$ yields

$$\langle \mathbf{v}_\alpha^\dagger(0, 0)\mathbf{v}_\alpha(-M, 0) \rangle \underset{M \to \infty}{\simeq} |\mathcal{A}_1(\alpha)|^2 \left( \frac{2\pi}{N} \right)^{4h_\alpha} e^{-4\pi h_\alpha M/N}\,. \tag{3.43}$$

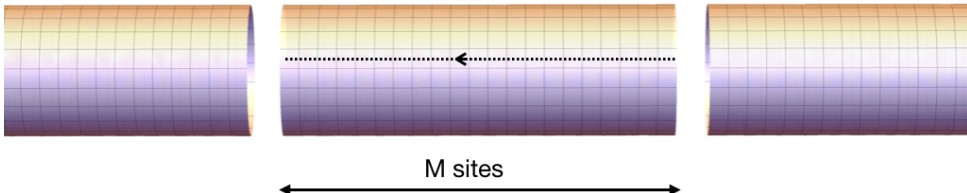

M sites

Figure 2: Inserting $\mathbf{v}_{-\alpha}(0,0)\mathbf{v}_\alpha(-M,0)$ amounts to twisting the periodic boundary conditions on the middle part of the cylinder. Within the transfer matrix formalism, the state at the edge of the leftmost cylinder is the untwisted ground state $|\psi_0(N)\rangle$, and likewise at the edge of the rightmost cylinder. Upon sending $M \to \infty$, the states at the edge of the middle cylinder are projected to the twisted ground state $|\psi_\alpha(N)\rangle$. Gluing back the cylinder yields the scalar product $|\langle\psi_0(N)|\psi_\alpha(N)\rangle|^2$.

In the transfer-matrix formalism, inserting $\mathbf{v}_{-\alpha}(0,0)\mathbf{v}_\alpha(-M,0)$ is exactly equivalent to setting *twisted* periodic boundary conditions

$$(\sigma^x_{N+1} \pm i\sigma^y_{N+1}) = e^{\pm i\alpha}(\sigma^x_1 \pm i\sigma^y_1), \qquad \sigma^z_{N+1} = \sigma^z_1, \tag{3.44}$$

on the portion of cylinder between $-M$ and 0. When $M \to +\infty$, this yields:

$$\langle\mathbf{v}^\dagger_\alpha(0,0)\mathbf{v}_\alpha(-M,0)\rangle \underset{M\to\infty}{\simeq} |\langle\psi_0(N)|\psi_\alpha(N)\rangle|^2 \times [\Lambda_\alpha(N)/\Lambda_0(N)]^M, \tag{3.45}$$

where $|\psi_\alpha(N)\rangle$ is the normalised ground state of the twisted XXZ chain, and $\Lambda_\alpha(N)$ is the associated eigenvalue of the transfer matrix (see Fig. 2). Comparing (3.43) and (3.45), we recover the asymptotic behaviour of eigenvalues $\lim_{N\to\infty} N\log[\Lambda_\alpha(N)/\Lambda_0(N)] = -4\pi h_\alpha$, and we obtain the normalising factor for $-\pi < \alpha < \pi$:

$$|\mathcal{A}_1(\alpha)|^2 = \lim_{N\to\infty}\left[\left(\frac{N}{2\pi}\right)^{4h_\alpha} \times |\langle\psi_0(N)|\psi_\alpha(N)\rangle|^2\right]. \tag{3.46}$$

For $(2m-1)\pi < \alpha < (2m+1)\pi$ with $m \in \mathbb{Z}$, the same argument holds, with $|\psi_\alpha\rangle$ replaced by the appropriate excited state of the Hamiltonian with twisted boundary conditions.

We now turn to the case $n > 1$, which involves the replicated XXZ model:

$$\mathcal{H}^{(n)}_{\mathrm{XXZ}} = -\frac{1}{2}\sum_{\mu=1}^n\sum_{j=1}^N\left[\sigma^x_{\mu,j}\sigma^x_{\mu,j+1} + \sigma^y_{\mu,j}\sigma^y_{\mu,j+1} + \Delta(\sigma^z_{\mu,j}\sigma^z_{\mu,j+1} - 1)\right], \tag{3.47}$$

where $\mu$ is the replica index, always considered modulo $n$ in the following. Let $|\psi_0(N)\rangle\!\rangle$ be the ground state of $\mathcal{H}^{(n)}_{\mathrm{XXZ}}$ with untwisted periodic boundary conditions $\sigma^a_{\mu,N+1} = \sigma^a_{\mu,1}$ for $a = x, y, z$. Since the $n$ copies of the XXZ model are totally decoupled in this setup, we have simply $|\psi_0(N)\rangle\!\rangle = |\psi_0(N)\rangle^{\otimes n}$.

Introducing the pair of operators $\mathbf{t}^\dagger_\alpha(0,0)\mathbf{t}_\alpha(-M,0)$ corresponds to changing the boundary conditions on the portion of cylinder between $-M$ and 0, to twisted periodic boundary conditions carrying a U(1) charge $\alpha$ *and* connecting (cyclically) the copies $\mu$ and $(\mu+1)$:

$$(\sigma^x_{\mu,N+1} \pm i\sigma^y_{\mu,N+1}) = e^{\pm i\alpha\delta_{\mu,n}}(\sigma^x_{\mu+1,1} \pm i\sigma^y_{\mu+1,1}), \qquad \sigma^z_{\mu,N+1} = \sigma^z_{\mu+1,1}. \tag{3.48}$$

Let $|\widetilde{\psi}_\alpha(N)\rangle\rangle$ be the corresponding groundstate. Using the same reasoning as for $\mathcal{A}_1(\alpha)$, we get

$$|\mathcal{A}_n(\alpha)|^2 = \lim_{N\to\infty}\left[\left(\frac{N}{2\pi}\right)^{4(h_\tau + h_\alpha/n)} \times |\langle\langle\psi_0(N)|\widetilde{\psi}_\alpha(N)\rangle\rangle|^2\right]. \tag{3.49}$$

Hence, the determination of $|\mathcal{A}_1(\alpha)|^2$ and $|\mathcal{A}_n(\alpha)|^2$ amounts to evaluating the leading behaviour of the scalar products $\langle\psi_0(N)|\psi_\alpha(N)\rangle$ and $\langle\langle\psi_0(N)|\widetilde{\psi}_\alpha(N)\rangle\rangle$ as the chain length $N$ tends to infinity.

Interestingly, for $\alpha = 0$ and $n = 2$, (3.49) relates the scaling behaviour of one-interval Rényi entropies to that of another measure of entanglement, namely the bipartite fidelity [38]. This can be easily extended to a multipartite fidelity for $n > 2$.

## 3.4 The free-fermion case

The spin-1/2 XXZ chain at $\Delta = 0$ (*a.k.a.* the XX spin chain) can be mapped via a Jordan-Wigner transformation to the tight-binding model

$$H = -\sum_{j=1}^{N}\left(c_{j+1}^\dagger c_j + c_j^\dagger c_{j+1}\right). \tag{3.50}$$

For such a non-interacting fermionic model, powerful techniques are available to evaluate both the full counting statistics [39] and the (symmetry-resolved) entanglement entropies [40–42]. The first key point is to express the (symmetry-resolved) entanglement spectrum in terms of the spectrum of the correlation matrix restricted to region $A$. This is known as the Peschel trick [43]. The second step is to realize that for a translation invariant tight-binding model, the correlation matrix is a Toeplitz matrix, for which the generalised Fisher-Hartwig conjecture can be brought to bear [21, 44–46].

The above-mentioned results are essentially exact lattice calculations, and as such they include all the finite-size corrections to the CFT prediction. [For this model with $g = \frac{1}{2}$, the central charge is $c = 1$ and the scaling dimension of a vertex operator $V_\alpha$ is $h_\alpha = \frac{1}{8}(\alpha/\pi)^2$.] In particular, the leading asymptotic corrections to the cumulants generating function obtained via the Fisher-Hartwig conjecture (equation (8) in [39]), should be encoded in the function $\mathcal{A}_n(\alpha)$. As discussed in Sec. 3.3, this function $\mathcal{A}_n(\alpha)$ can be obtained by simply evaluating the overlap between the twisted and untwisted ground states. We shall illustrate here the analytical computation of $\mathcal{A}_1(\alpha)$ for the XX chain through the latter approach. Technically, the computation is very similar to that of bipartite fidelity in the periodic XX chain – see [47].

In terms of the fermionic modes, the twisted boundary conditions (3.44) take the form

$$c_{N+1} = (-1)^{N_F - 1} e^{-i\alpha}c_1, \tag{3.51}$$

where $N_F$ is the total number of fermions, given by $N_F = N/2 - S^z$. The allowed eigenmodes in the sector of fixed $N_F$ are given by

$$\gamma_{\alpha,k} = \frac{1}{\sqrt{N}}\sum_{j=1}^{N}e^{ij\theta_{\alpha,k}}c_j, \qquad \theta_{\alpha,k} = \frac{2\pi}{N}\left(k - \frac{N_F + 1}{2} - \frac{\alpha}{2\pi}\right), \qquad k \in \{1, 2, \ldots, N\}. \tag{3.52}$$

The associated energies are $\varepsilon_{\alpha,k} = -2\cos\theta_{\alpha,k}$. For $N$ even and $-\pi < \alpha < \pi$, the ground state lies in the sector $N_F = N/2$. The corresponding left and right eigenvectors read:

$$\langle\psi_\alpha(N)| = \langle 0|\gamma_{\alpha,1}\gamma_{\alpha,2}\ldots\gamma_{\alpha,N/2}, \qquad |\psi_\alpha(N)\rangle = \gamma_{\alpha,1}^\dagger\gamma_{\alpha,2}^\dagger\ldots\gamma_{\alpha,N/2}^\dagger|0\rangle. \tag{3.53}$$

From Wick's theorem, the overlap is given by the determinant formula:

$$\langle\psi_0(N)|\psi_\alpha(N)\rangle = \det_{1\leqslant k,\ell\leqslant N/2} M_{k\ell}(\alpha), \qquad M_{k\ell}(\alpha) = \{\gamma_{0,k},\gamma_{\alpha,\ell}^\dagger\}. \tag{3.54}$$

The anticommutator is easily computed:

$$M_{k\ell}(\alpha) = \frac{\exp[\frac{i}{2}(\theta_{0,k}-\theta_{\alpha,\ell}+\alpha)]}{N}\frac{\sin\frac{\alpha}{2}}{\sin\frac{1}{2}(\theta_{0,k}-\theta_{\alpha,\ell})}. \tag{3.55}$$

Thus, ignoring an overall phase, the overlap can be expressed in terms of a Cauchy determinant:

$$|\langle\psi_0(N)|\psi_\alpha(N)\rangle| = \left(\frac{2\sin\frac{\alpha}{2}}{N}\right)^{N/2}\left|\det_{1\leqslant k,\ell\leqslant N/2}\left(\frac{1}{x_k-y_\ell}\right)\right|, \qquad x_k = e^{i\theta_{0,k}}, \quad y_\ell = e^{i\theta_{\alpha,\ell}}. \tag{3.56}$$

The Cauchy alternant formula

$$\det\left(\frac{1}{x_k-y_\ell}\right) = \frac{\prod_{k<\ell}(x_k-x_\ell)(y_k-y_\ell)}{\prod_{k,\ell}(x_k-y_\ell)}, \tag{3.57}$$

leads to the exact expression:

$$|\langle\psi_0(N)|\psi_\alpha(N)\rangle| = \left(\frac{\sin\frac{\alpha}{2}}{N\sin\frac{\alpha}{2N}}\right)^{N/2}\prod_{1\leqslant k<\ell\leqslant N/2}\frac{\sin^2\frac{\pi}{N}(k-\ell)}{\sin\frac{\pi}{N}\left(k-\ell-\frac{\alpha}{2\pi}\right)\sin\frac{\pi}{N}\left(k-\ell+\frac{\alpha}{2\pi}\right)}. \tag{3.58}$$

The asymptotic behaviour for large $N$ (see Appendix A) is given in terms of Barnes' G-function:

$$|\langle\psi_0(N)|\psi_\alpha(N)\rangle|\underset{N\to\infty}{\simeq}\left(\frac{\pi}{N}\right)^{2h_\alpha}G\left(1+\frac{\alpha}{2\pi}\right)G\left(1-\frac{\alpha}{2\pi}\right), \tag{3.59}$$

where $h_\alpha = \frac{1}{8}(\alpha/\pi)^2$, as expected from the CFT argument of the previous Section. Therefore, for the XX chain, we have the exact expression for the lattice prefactor:

$$|\mathcal{A}_1(\alpha)|^2 = 2^{-\frac{1}{2}(\alpha/\pi)^2}\left[G\left(1+\frac{\alpha}{2\pi}\right)G\left(1-\frac{\alpha}{2\pi}\right)\right]^2. \tag{3.60}$$

Inserting this into the generating function (3.15), we find

$$\log\widehat{Z}_1(\alpha)\simeq-\frac{\alpha^2}{2\pi^2}\log\left(\frac{L}{\pi a}\sin\frac{\pi\ell}{L}\right)+\log|\mathcal{A}_1|^2(\alpha), \tag{3.61}$$

and recover the leading correction as given in eq (8) of [39]. Computing the Taylor expansion of $|\mathcal{A}_1(\alpha)|^2$, we find

$$a_1 = -\frac{1+\gamma+\log 2}{2\pi^2}, \qquad a_2 = \frac{a_1^2}{2}-\frac{\zeta(3)}{16\pi^4}, \tag{3.62}$$

where $\gamma$ is the Euler-Mascheroni constant.

Generalising to the case $n > 1$ (see Appendix A), one gets

$$|\mathcal{A}_n(\alpha)|^2 = 2^{-4(h_\tau + h_\alpha/n)} \prod_{\substack{p=1-n \\ p=n-1 \bmod 2}}^{n-1} \left[ G\left(1 + \frac{p}{2n} + \frac{\alpha}{2\pi n}\right) G\left(1 + \frac{p}{2n} - \frac{\alpha}{2\pi n}\right) \right]^2 . \tag{3.63}$$

This is to be compared with the coefficient $\mathsf{a}_2$ in equation (3.21) of [41], namely

$$\mathsf{a}_2 = -\frac{1}{\pi i} \oint d\lambda \frac{df_n(\lambda, \alpha)}{d\lambda} \log\left(G(1 + \beta_\lambda) G(1 - \beta_\lambda)\right), \tag{3.64}$$

where

$$f_n(\lambda, \alpha) = \ln\left[\left(\frac{1+\lambda}{2}\right)^n e^{i\alpha} + \left(\frac{1-\lambda}{2}\right)^n\right], \qquad \beta_\lambda = \frac{1}{2\pi i} \ln\left(\frac{\lambda+1}{\lambda-1}\right). \tag{3.65}$$

Comparing our notations with those of [41], one should have

$$\mathsf{a}_2 = \log|\mathcal{A}_n(\alpha)|^2 + 4\left(h_\tau + \frac{h_\alpha}{n}\right)\log 2. \tag{3.66}$$

In order to compare with our results, we have to evaluate the above contour integral. First notice that the integrand is analytic at $\lambda = \infty$. This means that one can think of the contour as winding around infinity, thus avoiding the branch cut going from $\lambda = -1$ to $\lambda = +1$, and allowing for the residue theorem to be applied. Now the only poles come from the term $\frac{d}{d\lambda} f_n(\lambda, \alpha)$, which has simple poles (with residue 1) at every $\lambda$ such that

$$\beta_\lambda = -\frac{\alpha}{2\pi n} + \frac{1}{2} + \frac{1}{2n} \text{ modulo } \frac{1}{n}. \tag{3.67}$$

The residue theorem yields

$$\mathsf{a}_2 = 2 \sum_{\substack{p=1-n \\ p=n-1 \bmod 2}}^{n-1} \log\left[ G\left(1 + \frac{p}{2n} + \frac{\alpha}{2\pi n}\right) G\left(1 + \frac{p}{2n} - \frac{\alpha}{2\pi n}\right) \right]^2, \tag{3.68}$$

which is indeed our result (3.63).

Finally, we write down explicit formulas for the coefficients in the decomposition (3.38). We first obtain the Taylor expansion for the logarithm of $|\mathcal{A}_n(\alpha)/\mathcal{A}_n(0)|^2$:

$$\log\left|\frac{\mathcal{A}_n(\alpha)}{\mathcal{A}_n(0)}\right|^2 = \sum_{j=1}^{\infty} \frac{b_{n,j}}{(2j)!} \alpha^{2j}, \tag{3.69}$$

with

$$b_{n,j} = \frac{1}{2^{2j-2} n^{2j+1} \pi^{2j}} \sum_{k=1}^{\frac{n-1}{2}} k\left(\psi^{(2j-1)}(\tfrac{k}{n}) - \psi^{(2j-1)}(1 - \tfrac{k}{n})\right) - \begin{cases} \frac{1+\gamma+\log 2n}{2n\pi^2} & j=1, \\ \frac{\zeta(2j-1)}{2^{2j-1} nj\pi^{2j}} & j>1, \end{cases} \tag{3.70}$$

for $n$ odd and

$$b_{n,j} = \frac{1}{2^{2j-2} n^{2j+1} \pi^{2j}} \sum_{k=\frac{1}{2}, \frac{3}{2}, \cdots}^{\frac{n-1}{2}} k\left(\psi^{(2j-1)}(\tfrac{k}{n}) - \psi^{(2j-1)}(1 - \tfrac{k}{n})\right) - \begin{cases} \frac{1+\gamma+\log 8n}{2n\pi^2} & j=1, \\ \frac{(2^{2j-1}-1)\zeta(2j-1)}{2^{2j-1} nj\pi^{2j}} & j>1, \end{cases} \tag{3.71}$$

for $n$ even. Here $\psi^{(m)}(z)$ is the polygamma function of order $m$. The first coefficients $a_{n,j}$ are then given by

$$a_{n,1} = b_{n,1}, \qquad a_{n,2} = \tfrac{1}{2}\left[b_{n,2} + (b_{n,1})^2\right], \qquad a_{n,3} = \tfrac{1}{6}\left[b_{n,3} + 3b_{n,2}b_{n,1} + (b_{n,1})^3\right]. \quad (3.72)$$

In general, $a_{n,j}$ is expressed as a sum over the partitions $P = \{p_1^{n_1}, p_2^{n_2}, \ldots, p_m^{n_m}\}$ of length $j$:

$$a_{n,j} = \sum_{|P|=j} \prod_{i=1}^{m} \frac{(b_{n,p_i})^{n_i}}{(2p_i)! \, n_i!}. \qquad (3.73)$$

### 3.5 Numerical computations of lattice prefactors

We performed simple computations based on the exact diagonalisation of the XXZ Hamiltonian, to confirm our analysis of the finite-size behaviour of the quantities $\mathcal{A}_n(\alpha)$ and $\widehat{Z}_n(\alpha)$. For this purpose, we use a fitting formula for $\widehat{Z}_n(\alpha) = \langle \mathbf{t}_\alpha^\dagger(0,0)\mathbf{t}_\alpha(0,r)\rangle$, which only takes into account the three first primary operators contributing to the two-point function:

$$\widehat{Z}_n(\alpha) \simeq \frac{|\mathcal{A}_n(\alpha)|^2}{w^{4(h_\tau + h_\alpha/n)}} + (-1)^r \left[\frac{|\mathcal{A}_n(\alpha - 2\pi)|^2}{w^{4(h_\tau + h_{\alpha-2\pi}/n)}} + \frac{|\mathcal{A}_n(\alpha + 2\pi)|^2}{w^{4(h_\tau + h_{\alpha+2\pi}/n)}}\right], \qquad w = \frac{N}{\pi}\sin\frac{\pi r}{N}. \quad (3.74)$$

For the XX spin chain (see Fig. 3 a,b,c), since we have an analytical determination of the scalar product $\langle\!\langle\psi_0|\widetilde{\psi}_\alpha\rangle\!\rangle$, we can compare the exact formula (3.63) with the prefactor $|\mathcal{A}_n(\alpha)|^2$ in (3.74). In practice, to compute $\widehat{Z}_n(\alpha)$, we first obtain the ground state $|\psi_0(N)\rangle$ by a Lanczos algorithm, then construct and diagonalise the reduced density matrices $\rho_A(q)$, and finally perform the trace (2.11). Quite remarkably, the agreement with (3.63) is excellent, even for small system sizes.

For the XXZ spin chain with a generic anisotropy parameter $\Delta$, it is also useful to evaluate the coefficients $|\mathcal{A}_n(\alpha)|^2$ from the numerical computation of overlaps (3.46) and (3.49). Due to memory usage, we restricted this numerical study to the case $n = 1$. In Fig. 4 we show the comparison between the numerical determinations of $|\mathcal{A}_1(\alpha)|^2$ for $\Delta = 0.4$, from the overlap (3.46) and from the two-point function (3.74). In Fig. 5 we show $|\mathcal{A}_n(\alpha)|^2$ as a function of $\alpha$, for various values of $\Delta$.

The above results show that the coefficients $|\mathcal{A}_n(\alpha)|^2$ can be obtained numerically with very moderate computing time and space, on the basis of an exact diagonalisation of the Hamiltonian for system sizes of order 10-20 sites.

## 4 The critical clock-spin chain

### 4.1 The clock Hamiltonian

The critical $\mathbb{Z}_p$ clock model [25] on the square lattice is a simple example of an integrable critical model with an internal discrete symmetry – the cyclic group $\mathbb{Z}_p$. The anisotropic limit of the transfer matrix yields the critical 1d Hamiltonian, acting on the Hilbert space $(\mathbb{C}^p)^{\otimes N}$:

$$\mathcal{H}_{\text{clock}} = -\sum_{j=1}^{N}\sum_{k=1}^{p-1} \frac{1}{\sin(\pi k/p)}\left[(Z_j Z_{j+1}^\dagger)^k + (X_j)^k\right], \qquad (4.1)$$

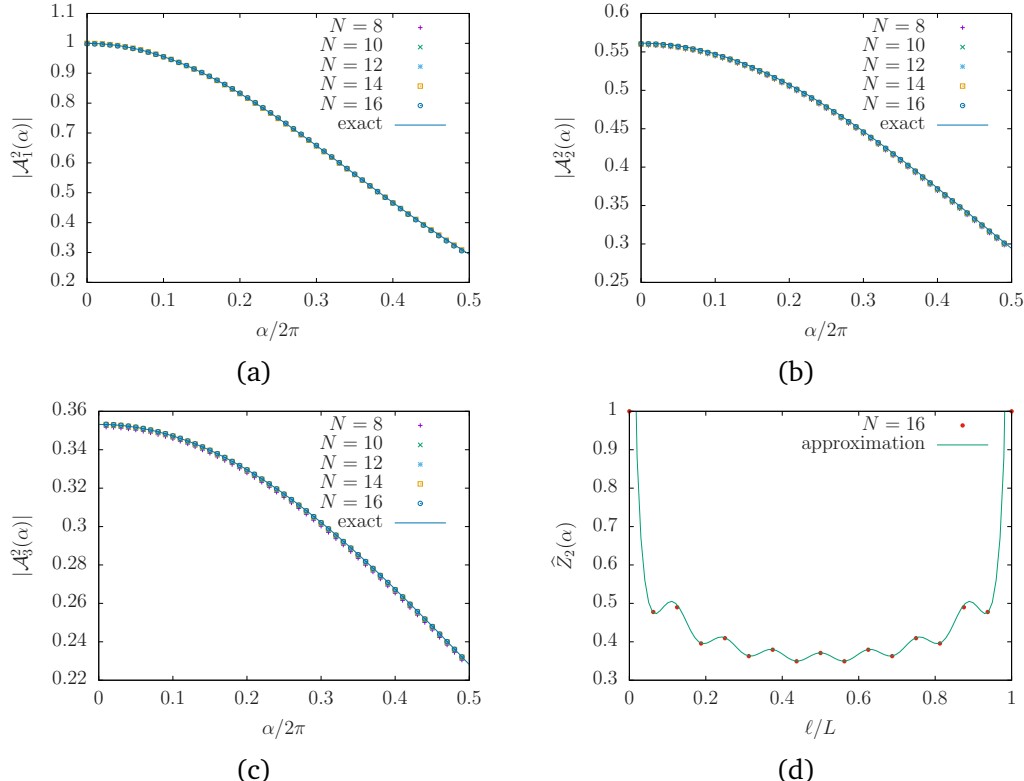

Figure 3: The functions $\mathcal{A}_n(\alpha)$ and $\widehat{Z}_n(\alpha)$ for the periodic XX chain. (a,b,c) The prefactors $|\mathcal{A}_n(\alpha)|^2$, obtained from the numerical computation of the two-point function $\langle \mathbf{t}_\alpha^\dagger \mathbf{t}_\alpha \rangle$ for $n = 1, 2, 3$ on a cylinder, are compared with the exact formula (3.63). (d) The numerical two-point function $\langle \mathbf{t}_\alpha^\dagger \mathbf{t}_\alpha \rangle$ for $n = 2$ and $N = 16$ is plotted against the approximation (3.74), with coefficients $|\mathcal{A}_2(\alpha)|^2$ taken from the exact formula (3.63).

with periodic boundary conditions $Z_{N+1} = Z_1$, and where the local matrices are defined by

$$X_j = \mathbf{1} \otimes \cdots \otimes \mathbf{1} \otimes \underset{j-\text{th}}{X} \otimes \mathbf{1} \otimes \cdots \otimes \mathbf{1}, \tag{4.2}$$

$$Z_j = \mathbf{1} \otimes \cdots \otimes \mathbf{1} \otimes \underset{j-\text{th}}{Z} \otimes \mathbf{1} \otimes \cdots \otimes \mathbf{1}. \tag{4.3}$$

The elementary matrices $X$ and $Z$ are defined by their matrix elements

$$X_{ab} = \begin{cases} 1 & \text{if } a = b + 1 \mod p \\ 0 & \text{otherwise,} \end{cases} \qquad Z_{ab} = \exp(2i\pi a/p)\,\delta_{ab}. \tag{4.4}$$

In the scaling limit, this model is described by the $\mathbb{Z}_p$ parafermionic CFT [48].

The overall rotation of spins $\mathcal{X}$ is a local $\mathbb{Z}_p$ conserved charge

$$\mathcal{X} = \prod_{j=1}^{N} X_j, \qquad [\mathcal{X}, \mathcal{H}_{\text{clock}}] = 0. \tag{4.5}$$

If we partition the system into two subsystems $A = \{1, \ldots, r\}$ and $B = \{r + 1, \ldots, N\}$, we can

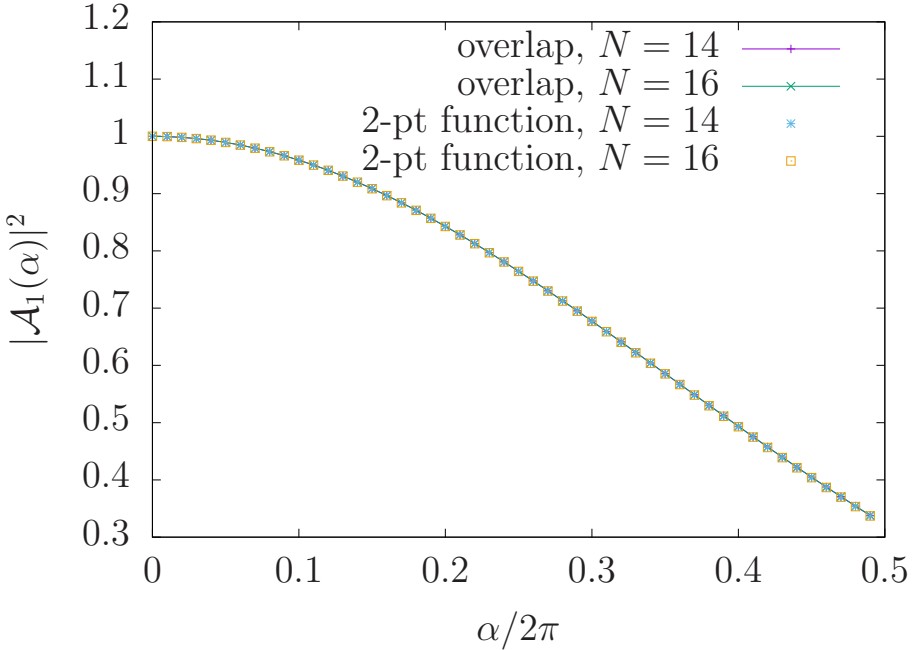

Figure 4: The coefficient $\mathcal{A}_1(\alpha)$ as a function of $\alpha$, for the XXZ spin chain with $\Delta = 0.4$ and system sizes of $N = 14, 16$ sites. The quantity $|\mathcal{A}_1(\alpha)|^2$ is obtained from the overlap (3.46), or from the two-point function (3.74). The agreement between the two determinations is very good.

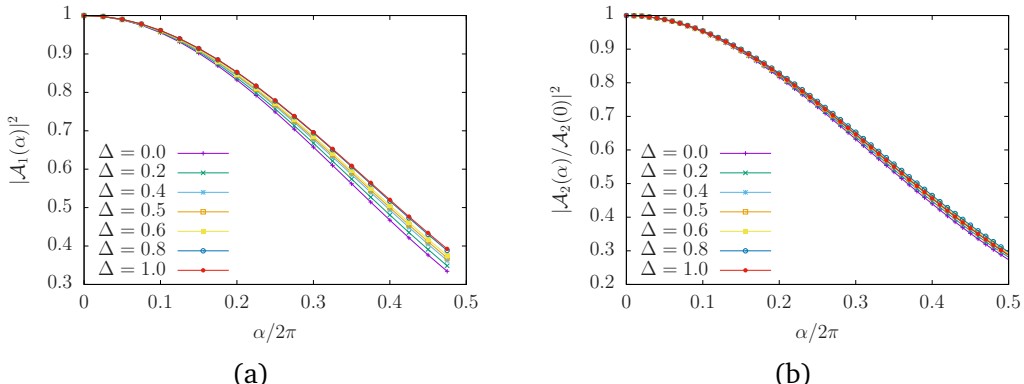

Figure 5: The coefficient $\mathcal{A}_1(\alpha)$ as a function of $\alpha$, for the XXZ spin chain with various values of $\Delta$. (a) The quantity $|\mathcal{A}_1(\alpha)|^2$ is obtained from the overlap (3.46), for a system of size $N = 16$ sites. (b) The quantity $|\mathcal{A}_2(\alpha)|^2$ is obtained from the two-point function $\widehat{Z}_2(\alpha)$ and the fit (3.74), for a system of size $N = 16$ sites. A small dependence on $\Delta$ can be observed, which cannot be attributed to finite-size effects.

define the $\mathbb{Z}_p$ charge of $A$ as

$$\mathcal{X}_A = \prod_{j=1}^{r} X_j = e^{2i\pi Q_A/p}, \tag{4.6}$$

with eigenvalues of the form $\exp(2i\pi q/p)$, where $q \in \mathbb{Z}_p$. We define the generating function of resolved entropies as

$$\widehat{Z}_n(\alpha) = \mathrm{Tr}_A\left(\mathcal{X}_A^{\alpha} \rho_A^n\right), \tag{4.7}$$

where $\alpha \in \mathbb{Z}_p$, and $\rho_A$ is the reduced density matrix. One recovers the resolved partition function as

$$Z_n(q) = \frac{1}{p} \sum_{\alpha=0}^{p-1} \exp(-2i\pi\alpha q/p)\widehat{Z}_n(\alpha). \tag{4.8}$$

## 4.2 Probability distribution of the partial charge

The probability distribution for the partial charge is given by

$$p(q) = Z_1(q) = \frac{1}{p} \sum_{\alpha=0}^{p-1} \cos(2\pi\alpha q/p)\widehat{Z}_1(\alpha). \tag{4.9}$$

The generating function $\widehat{Z}_A^{(1)}(\alpha)$ is simply the two-point function of the disorder operator $\mu_\alpha$ in the clock model on the infinite cylinder (see [48]):

$$\widehat{Z}_1(\alpha) = \langle \mu_\alpha^\dagger(0,0)\mu_\alpha(0,r)\rangle. \tag{4.10}$$

By construction, we have $\widehat{Z}_1(0) = 1$. In the scaling limit, $\mu_\alpha$ becomes a linear combination of scaling operators in the sector of spin $\mathbb{Z}_p$ charge zero and dual $\mathbb{Z}_p$ charge $\alpha$ (see [48]). For a given integer $\alpha$ in the range $\{1, 2, \ldots N-1\}$, the most relevant operator in this sector has the same dimension as the $\sigma_\alpha$ operator:

$$h_\alpha = \frac{\alpha(p-\alpha)}{2p(p+2)}. \tag{4.11}$$

The dimension of the elementary spin operator $\sigma_1 = \sigma$ is denoted

$$h_\sigma = \frac{p-1}{2p(p+2)}. \tag{4.12}$$

For instance, we have $h_\sigma = 1/16$ for the Ising model, and $h_\sigma = 1/15$ for the three-state Potts model. Hence, the two-point function is of the form

$$\widehat{Z}_1(\alpha) \simeq |\mathcal{A}_1(\alpha)|^2 \left(\frac{L}{\pi a} \sin \frac{\pi \ell}{L}\right)^{-4h_\alpha}, \qquad \alpha \in \mathbb{Z}_p, \tag{4.13}$$

where $\mathcal{A}_1(\alpha)$ is the non-universal normalisation factor of the lattice operator $\mu_\alpha$. Since the model is invariant under spin-reversal, this factor satisfies $\mathcal{A}_1(-\alpha) = \mathcal{A}_1(\alpha)^*$. Hence, the dominant behaviour of $p_q$ is

$$p_q \simeq \frac{1}{p}\left[1 + 2\cos\frac{2\pi q}{p} \times |\mathcal{A}_1(1)|^2 \left(\frac{L}{\pi a} \sin \frac{\pi \ell}{L}\right)^{-4h_\sigma}\right], \tag{4.14}$$

where $h_\sigma$ is the spin conformal dimension (4.12).

### 4.3  Charge-resolved entropies

Like in the XXZ case, for $n > 1$ we can view the generating function $\widehat{Z}_n(\alpha)$ as the two-point function of a composite twist operator $\mathbf{t}_\alpha = \mathbf{t} \cdot \mu_\alpha$ in the $\mathbb{Z}_n$ orbifold of the lattice model:

$$\widehat{Z}_n(\alpha) = \left\langle \mathbf{t}_\alpha^\dagger(0,0)\mathbf{t}_\alpha(0,r) \right\rangle \simeq |\mathcal{A}_n(\alpha)|^2 \left( \frac{L}{\pi a} \sin \frac{\pi \ell}{L} \right)^{-4(h_\tau + h_\alpha/n)}, \tag{4.15}$$

where $\mathcal{A}_n(\alpha)$ is the normalisation factor of the lattice operator $t_\alpha$. We write

$$\frac{Z_n(q)}{\widehat{Z}_n(0)} = \frac{1}{p} \sum_{\alpha=0}^{p-1} \cos \frac{2\pi q \alpha}{p} \frac{\widehat{Z}_n(\alpha)}{\widehat{Z}_n(0)} \simeq \frac{1}{p} \left[ 1 + 2\cos \frac{2\pi q}{p} \times \left| \frac{\mathcal{A}_n(1)}{\mathcal{A}_n(0)} \right|^2 \left( \frac{L}{\pi a} \sin \frac{\pi \ell}{L} \right)^{-4h_\sigma/n} \right]. \tag{4.16}$$

For $p = 2$, we recover the scaling behaviour found in previous works on the XY chain [6, 42] – our analysis explains the presence of constant prefactors observed in these works.

The behaviour of the charge-resolved entropies is:

$$S_n(q) = S_n - \log p + 2\cos\left( \frac{2\pi q}{p} \right) \times c_n(L, \ell), \tag{4.17}$$

where the dominant correction terms are of the form

$$c_n(L, \ell) \simeq \frac{1}{1-n} \left[ \left| \frac{\mathcal{A}_n(1)}{\mathcal{A}_n(0)} \right|^2 \left( \frac{L}{\pi a} \sin \frac{\pi \ell}{L} \right)^{-4h_\sigma/n} - n \left| \frac{\mathcal{A}_1(1)}{\mathcal{A}_1(0)} \right|^2 \left( \frac{L}{\pi a} \sin \frac{\pi \ell}{L} \right)^{-4h_\sigma} \right]. \tag{4.18}$$

### 4.4  Lattice prefactors

Like in the XXZ case (see Sec. 3.3), the coefficients $\mathcal{A}_n(\alpha)$ can be obtained from the asymptotic behaviour of overlaps between groundstates of the replicated clock-model Hamiltonian with twisted and untwisted periodic boundary conditions:

$$|\mathcal{A}_n(\alpha)|^2 = \lim_{N\to\infty} \left[ \left( \frac{N}{2\pi} \right)^{4(h_\tau + h_\alpha/n)} \times |\langle\!\langle \psi_0(N) | \widetilde{\psi}_\alpha(N) \rangle\!\rangle|^2 \right]. \tag{4.19}$$

In the above expression, we have considered the replicated Hamiltonian

$$\mathcal{H}_{\text{clock}}^{(n)} = -\sum_{\mu=1}^{n} \sum_{j=1}^{N} \sum_{k=1}^{p-1} \frac{1}{\sin(\pi k/p)} \left[ (Z_{\mu,j} Z_{\mu,j+1}^\dagger)^k + (X_{\mu,j})^k \right], \tag{4.20}$$

with untwisted periodic conditions $Z_{\mu,N+1} = Z_{\mu,1}$ for the state $|\psi_0(N)\rangle\!\rangle$, and twisted periodic boundary conditions $Z_{\mu,N+1} = e^{2i\pi\alpha\delta_{\mu,n}/p} Z_{\mu+1,1}$ for $|\widetilde{\psi}_\alpha(N)\rangle\!\rangle$.

In particular, for $p = 2$ where the model is equivalent to a special point of the (free-fermionic) XY spin chain, it should be possible to compute the twisted overlaps.

## 5  Conclusion

In this article we have analysed the leading finite-size corrections to the symmetry-resolved (Rényi) entanglement entropy. For a discrete $\mathbb{Z}_p$ symmetry, these corrections decay algebraically with system size, with a universal exponent controlled by the conformal dimension of

the most relevant composite twist field. In the case of U(1) symmetry however, the finite-size scaling exhibits very large logarithmic corrections. They originate from the normalization of the lattice composite twist field used in the Replica trick, and are a priori non universal. We have shown that these normalizations are related to the overlap between the untwisted and the twisted ground state of the model. As a check, we computed these overlaps for a free fermionic system and recovered the leading logarithmic corrections obtained in [39, 41].

For the critical XXZ chain with $\Delta \neq 0$, the determination of lattice prefactors through the method described in Sec. 3.3 would involve an asymptotic analysis of overlaps $\langle \psi_0(N)|\psi_\alpha(N)\rangle$ and $\langle\!\langle \psi_0(N)|\widetilde{\psi}_\alpha(N)\rangle\!\rangle$ for fixed $\alpha$ and large $N$. For $\langle \psi_0(N)|\psi_\alpha(N)\rangle$, despite the existence of a determinantal formula [49,50] in terms of the roots of Bethe Ansatz equations, this asymptotic analysis in the critical regime $|\Delta| < 1$ remains a non-trivial problem. For $\langle\!\langle \psi_0(N)|\widetilde{\psi}_\alpha(N)\rangle\!\rangle$, a first important step would be to obtain a determinantal formula analogous to [49, 50], and then perform its asymptotic analysis. We hope to tackle these problems in future work.

## Acknowledgments

The authors are thankful to P. Calabrese, M. Goldstein, E. Sela and G. Parez for their comments on the manuscript.

## A  Asymptotics of scalar products for the XX chain

In this appendix, we compute the overlap $|\langle\!\langle \psi_0(N)|\widetilde{\psi}_\alpha(N)\rangle\!\rangle|$ for arbitrary values of $n$ in the XX chain. To do this, we shall use the fact that $|\widetilde{\psi}_\alpha(N)\rangle\!\rangle = |\psi_\alpha(nN)\rangle$. Thus we consider a chain of size $nN$, and compute $|\langle\!\langle \psi_0(N)|\widetilde{\psi}_\alpha(N)\rangle\!\rangle|$ as the overlap between the ground state of the total system, and the tensor product of ground states of $n$ adjacent subsystems of size $N$.

The fermionic operators for the full system of length $Nn$ are

$$\gamma_k = \frac{1}{\sqrt{Nn}} \sum_{j=1}^{Nn} e^{ij\theta_k} c_j, \qquad \gamma_k^\dagger = \frac{1}{\sqrt{Nn}} \sum_{j=1}^{Nn} e^{-ij\theta_k} c_j^\dagger, \qquad \theta_k = \frac{2\pi}{Nn}\left(k - \frac{Nn}{4} - \frac{1}{2} - \frac{\alpha}{2\pi}\right). \quad \text{(A.1)}$$

The groundstate eigenvector of $H$ for even values of $N$ is

$$|\widetilde{\psi}_\alpha(nN)\rangle\!\rangle = \gamma_1^\dagger \gamma_2^\dagger \cdots \gamma_{\frac{Nn}{2}}^\dagger |0\rangle. \quad \text{(A.2)}$$

We introduce the fermionic operators specific to each subsystem,

$$\gamma_k^{(\mu)} = \frac{1}{\sqrt{N}} \sum_{j=1+(\mu-1)N}^{\mu N} e^{i(j-(\mu-1)N)\eta_k} c_j, \qquad \eta_k = \frac{2\pi}{N}\left(k - \frac{N}{4} - \frac{1}{2}\right), \qquad \mu = 1,\ldots,n. \quad \text{(A.3)}$$

The state $\langle\!\langle \psi_0(N)|$ is then given by

$$\langle\!\langle \psi_0(N)| = \langle 0|\left(\gamma_1^{(1)}\cdots\gamma_{\frac{N}{2}}^{(1)}\right) \cdot \left(\gamma_1^{(2)}\cdots\gamma_{\frac{N}{2}}^{(2)}\right) \cdots \left(\gamma_1^{(n)}\cdots\gamma_{\frac{N}{2}}^{(n)}\right). \quad \text{(A.4)}$$

The fermionic operators satisfy the anticommutation relations

$$\{\gamma_k^{(\mu)}, \gamma_\ell^\dagger\} = e^{-i\theta_\ell(\mu-1)N} \frac{e^{i(\eta_k-\theta_\ell)N} - 1}{2i \sin\frac{1}{2}(\eta_k - \theta_\ell)}. \quad \text{(A.5)}$$

Using Wick's theorem, we write the overlap as

$$\langle\!\langle\psi_0(N)|\widetilde{\psi}_\alpha(N)\rangle\!\rangle = \det_{k,\ell=1}^{Nn/2} M_{kl}, \tag{A.6}$$

where

$$M_{k\ell} = \{\gamma^{(\mu)}_{k-(\mu-1)N}, \gamma^{\dagger}_\ell\}, \qquad (\mu-1)N < k \leqslant \mu N, \qquad \mu = 1,\ldots,n. \tag{A.7}$$

Evaluating the determinant of $M$ is possible because only the first factor on the right side of (A.5) depends on $\mu$.

We proceed by making a change of basis for the fermions $\gamma^{(\mu)}_k$. The fermionic operators in the new basis are the discrete Fourier transforms of the original operators $\gamma^{(\mu)}_k$ over the $n$ subsystems. The matrix $\mathcal{U}$ of the change of basis is a block matrix defined as

$$\mathcal{U}_{ij} = \omega^{ij}\beta^j \mathbf{1}_{\frac{N}{2}}, \qquad \omega = e^{2\pi i/n}, \qquad \beta = \begin{cases} e^{i(\pi-\alpha)/n} & n \text{ even}, \\ (-1)^{Nn/2-1}e^{-i\alpha/n} & n \text{ odd}, \end{cases} \tag{A.8}$$

where $i,j = 0,\ldots,n-1$, and $\mathbf{1}_p$ is the identity matrix of size $p$. After a reordering of its columns, the matrix $\mathcal{U}M$ is block-diagonal. Using

$$|\det\mathcal{U}| = n^{Nn/4}, \tag{A.9}$$

we simplify the expression for the overlap and find

$$|\langle\!\langle\psi_0(N)|\widetilde{\psi}_\alpha(N)\rangle\!\rangle| = \left|(2N)^{-\frac{Nn}{2}}(2\cos(\tfrac{\alpha}{2}+\tfrac{n\pi}{2}))^{\frac{N}{2}}\prod_{p=0}^{n-1}\det M^{(p)}\right|, \tag{A.10}$$

with

$$M^{(p)}_{k\ell} = \frac{1}{\sin\frac{1}{2}(\eta_k - \theta_{n\ell-p})}, \qquad k,\ell = 1,\ldots,\tfrac{N}{2}. \tag{A.11}$$

Each of the determinants on the right-hand side can be evaluated using Cauchy's determinant formula:

$$|\det M^{(p)}| = \left|\frac{\prod_{1\leqslant k<\ell\leqslant N/2}\sin^2\left(\frac{(k-\ell)\pi}{N}\right)}{\prod_{k,\ell=1}^{N/2}\sin\left(\frac{(k-\ell)\pi}{N}+\frac{\alpha}{2N}-\frac{\pi(n-2p-1)}{2Nn}\right)}\right|. \tag{A.12}$$

We have thus obtained an explicit formula for $|\langle\!\langle\psi_0(N)|\widetilde{\psi}_\alpha(N)\rangle\!\rangle|$ in terms of products of trigonometric functions.

To compute the leading terms of its large-$N$ expansion, we define the functions

$$X(a,N) = \sum_{k=0}^{N/2}\log\operatorname{sinc}\left[\frac{\pi}{N}(k+a)\right], \qquad Y(a,N) = \sum_{k=0}^{N/2}\sum_{\ell=0}^{N/2}\log\operatorname{sinc}\left[\frac{\pi}{N}\left(k-\ell+a\right)\right]. \tag{A.13}$$

We rewrite the sine functions in (A.12) in terms of sinc functions. Each of the products is then doubled. For example, we have

$$\prod_{k=1}^{N/2}\prod_{\ell=k+1}^{N/2}\sin^2\left(\frac{(k-\ell)\pi}{N}\right) = \left(\frac{\pi}{N}\right)^{N(N-2)/4}\prod_{k=1}^{N/2}\prod_{\ell=k+1}^{N/2}(k-\ell)^2\operatorname{sinc}^2\left(\frac{(k-\ell)\pi}{N}\right). \tag{A.14}$$

The first product involves only the arguments of the sine functions and is easily rewritten in terms of the Barnes G-function:

$$\prod_{k=1}^{N/2}\prod_{\ell=k+1}^{N/2}(\ell-k) = \prod_{k=1}^{N/2}\prod_{\ell=k+1}^{N/2}\frac{\Gamma(\ell-k+1)}{\Gamma(\ell-k)} = \prod_{k=1}^{N/2}\Gamma(\tfrac{N}{2}-k+1) = \prod_{k=1}^{N/2}\Gamma(k) = G(\tfrac{N}{2}+1). \tag{A.15}$$

The second product involves the sinc functions and is expressed in terms of the functions $X$ and $Y$:

$$\prod_{k=1}^{N/2}\prod_{\ell=k+1}^{N/2}\text{sinc}^2\left(\frac{(k-\ell)\pi}{N}\right) = \prod_{k,\ell=1}^{N/2}\text{sinc}\left(\frac{(k-\ell)\pi}{N}\right) = \frac{e^{Y(0,N)}}{e^{2X(0,N)}}. \tag{A.16}$$

Repeating the same argument with the denominator of (A.12), we find after some algebra

$$|\langle\!\langle\psi_0(N)|\widetilde{\psi}_\alpha(N)\rangle\!\rangle| = \prod_{p=0}^{n-1}\left[\frac{G^2(\frac{N}{2}+1)G(\frac{1}{2}+\frac{\alpha}{2\pi n}+\frac{2p+1}{2n})G(\frac{3}{2}-\frac{\alpha}{2\pi n}-\frac{2p+1}{2n})}{G(\frac{N}{2}+\frac{1}{2}+\frac{\alpha}{2\pi n}+\frac{2p+1}{2n})G(\frac{N}{2}+\frac{3}{2}-\frac{\alpha}{2\pi n}-\frac{2p+1}{2n})}\right. \tag{A.17}$$

$$\left.\times \frac{e^{X(\frac{\alpha}{2\pi n}+\frac{2p+1}{2n}-\frac{1}{2},N)}e^{X(-\frac{\alpha}{2\pi n}-\frac{2p+1}{2n}+\frac{1}{2},N)}e^{Y(0,N)}}{\text{sinc}\left(\frac{\alpha}{2Nn}-\frac{\pi(n-2p-1)}{2n}\right)e^{Y(\frac{\alpha}{2\pi n}+\frac{2p+1}{2n}-\frac{1}{2},N)}e^{2X(0,N)}}\right].$$

The asymptotic expansions of the functions $X$ and $Y$ are obtained from the Euler-Maclaurin formula:

$$X(a,N) = \frac{N}{2}(1-\log\pi)-\log\left(\frac{\pi}{2}\right)\left(a+\frac{1}{2}\right)+\mathcal{O}\left(N^{-1}\right), \tag{A.18}$$

$$Y(a,N) = \frac{N^2\mathcal{J}}{4}+N(1-\log\pi)-\log\left(\frac{\pi}{2}\right)\left(a^2+\frac{5}{6}\right)+\mathcal{O}\left(N^{-1}\right), \tag{A.19}$$

where

$$\mathcal{J} = 2\int_0^1 dz\,(1-z)\log\text{sinc}(\tfrac{\pi z}{2}) = \frac{1}{2}\left(3-2\log\pi-\frac{7}{\pi^2}\zeta(3)\right). \tag{A.20}$$

Likewise, the known asymptotics of the Barnes G-function is

$$\log G(z) = \left(\frac{(z-1)^2}{2}-\frac{1}{12}\right)\log(z-1)-\frac{3(z-1)^2}{4}+\frac{z-1}{2}\log(2\pi)+\frac{1}{12}-\log A+\mathcal{O}(z^{-1}). \tag{A.21}$$

This allows us to compute the two leading orders in the large-$N$ expansion of the logarithm of the overlap:

$$\log|\langle\!\langle\psi_0(N)|\widetilde{\psi}_\alpha(N)\rangle\!\rangle| = \frac{(1-n^2)-3(\alpha/\pi)^2}{12n}\log\left(\frac{N}{\pi}\right)$$

$$+\sum_{p=0}^{n-1}\log\left(G(\tfrac{1}{2}+\tfrac{\alpha}{2\pi n}+\tfrac{2p+1}{2n})G(\tfrac{3}{2}-\tfrac{\alpha}{2\pi n}-\tfrac{2p+1}{2n})\right)+O(N^{-1}). \tag{A.22}$$

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
