# Peer review of "Finite-size corrections in critical symmetry-resolved entanglement"

_SciPost Physics, doi:SciPost Phys. 10, 054 (2021)_

## Round 2 · Referee Report · Anonymous · 2020-12-18

Strengths

1-very well explained and written
2- extensive and careful study using state-of-the-art techniques
3- excellent verification of the results

Report

In this paper the authors study the finite-size corrections to the symmetry resolved entanglement entropies in systems described by conformal field theories. In the
scaling limit of large systems the entanglement entropies exhibit equipartition, i.e, they become independent on the symmetry sector. The authors show that the subleading finite-size corrections depend on the nature of the symmetry. For continuous symmetry they find that the corrections are logarithmic, whereas for discrete symmetry they are algebraic. The authors carefully check their findings numerically.

This is a very interesting paper which provides top-quality results on a timely topic. The paper provides an in-depth understanding of the origin of the finite-size corrections to the symmetry-resolved entanglement entropies. Moreover, to my knowledge the authors are the first to highlight the difference between continuous and discrete symmetries.

I have only one important observation:

1) The fact that the symmetry-resolved entanglement entropies exhibit
logarithmic corrections remains true also in the presence of disorder. This
has been studied recently in

Phys. Rev. B 102, 014455 (2020)

the authors could mention this paper.

Requested changes

1-some of the qualitative behaviors described are not new.

  • validity: top
  • significance: high
  • originality: high
  • clarity: high
  • formatting: perfect
  • grammar: excellent

Author:  Yacine Ikhlef  on 2021-01-18  [id 1160]

(in reply to Report 1 on 2020-12-18)

We thank the referee for the report and the suggestion for an additional citation. We agree this reference is relevant, and we shall include it to the manuscript.

---

## Round 2 · Referee Report · Anonymous · 2021-1-1

Strengths

1- Very well-written paper, easy to follow
2- Nice CFT calculations, complemented by thorough numerical checks

Report

In this paper, the authors study the finite-size corrections to symmetry-resolved entanglement in 1+1d CFTs. They find that corrections to entropy equipartition depend crucially on whether the symmetry group is discrete or continuous: for discrete symmetries, the corrections are algebraic, while in the case of a U(1) symmetry, the corrections decay logarithmically with system size. They also identify the prefactors of these corrections with some overlaps that can be computed numerically or analytically in some cases.

This work contains interesting results in a timely area of research. The manuscript is very well written, and easy to follow. I do not have any particular suggestion to improve the presentation, and I strongly recommend publication in SciPost as is.

---

## Round 3 · Author Response

We thank the referees for their reports and the suggestion for an additional citation. We agree this reference is relevant, and we have included it to the resubmitted manuscript.

---

## Round 3 · List of Changes

We have added the reference to Phys. Rev. B 102, 014455 (2020) , it is cited as [17] from the second paragraph of the Introduction section.

---

## Editorial Decision

published